# Genome Characterization of Carbapenem-Resistant Hypervirulent *Klebsiella pneumoniae* Strains, Carrying Hybrid Resistance-Virulence IncHI1B/FIB Plasmids, Isolated from an Egyptian Pediatric ICU

**DOI:** 10.3390/microorganisms13051058

**Published:** 2025-05-01

**Authors:** Heba A. Hammad, Radwa Abdelwahab, Douglas F. Browning, Sherine A. Aly

**Affiliations:** 1Department of Medical Microbiology and Immunology, Faculty of Medicine, Assiut University, Assiut 71515, Egypt; hebaali@aun.edu.eg (H.A.H.); radwa.wahab418@gmail.com (R.A.); 2College of Health & Life Sciences, Aston University, Aston Triangle, Birmingham B4 7ET, UK

**Keywords:** *Klebsiella pneumoniae*, antibiotic resistance, virulence, hvKP, whole-genome sequencing

## Abstract

Despite the increased reporting of Carbapenem-resistant hypervirulent *Klebsiella pneumoniae* (CR-hvKp) in Egypt, there is a paucity of information regarding the molecular characteristics of such strains. Herein, we present the genome sequence of two CR-hvKp strains, K22 and K45, which were isolated from VAP (ventilator-associated-pneumonia) patients admitted to pediatric ICU at Assiut University Children’s Hospital, Egypt. K22 and K45 isolates were subjected to antimicrobial susceptibility testing and whole-genome sequencing. Genomic analysis was performed to characterize each strain, determining their plasmids, antimicrobial resistance (AMR) genes, and virulence determinants. K22 possessed an extensive drug resistance phenotype (XDR), whilst K45 exhibited a multidrug resistance phenotype (MDR), with genome sequencing revealing the presence of a diverse array of AMR genes. Both strains were resistant to the carbapenem antibiotic imipenem, carrying the OXA-48 carbapenemase, with K22 additionally possessing an NDM-1 carbapenemase. Each strain was considered high-risk, with K22 and K45 respectively belonging to sequence types ST383 and ST14 and possessing virulence genes implicated in hypervirulence (e.g., *iucABCD-iutA* and *rmpA*). Importantly, both strains carried multiple plasmid replicons, including an AMR/virulence IncHI1B/FIB hybrid plasmid and MDR IncL/M plasmids. This report highlights the critical role of plasmids in the evolution of virulent *K. pneumoniae* strains and suggests the circulation of an IncHI1B/FIB hybrid plasmid, simultaneously disseminating AMR and hypervirulence, amongst *K. pneumoniae* strains within Assiut University Children’s Hospital.

## 1. Introduction

*Klebsiella pneumoniae* is a Gram-negative, encapsulated bacterium that colonizes the human gut and frequently causes healthcare-associated infections [1,2]. Hypervirulent *K. pneumoniae* (hvKp) is a distinct pathotype first recognized in the 1980s, that evolved through the acquisition of potent virulence determinants. HvKp signaled the onset of a concerning public health threat owing to its ability to produce invasive infections even in healthy immunocompetent adults [1]. Nevertheless, hospital-acquired hvKp infections has been increasingly reported worldwide [3,4,5]. Generally, community-acquired hvKp infections result in pyogenic liver abscess, bacteremia, and pneumonia, while health care-associated hvKp infections tend to lead to ventilator-associated pneumonia (VAP), catheter-associated urinary tract infection, and surgical site infections [2]. Furthermore, hvKp strains can colonize healthy individuals, with a propensity for dissemination to otherwise healthy individuals [6].

Initially, hvKp strains were susceptible to many different classes of antibiotics. Along with the reported worldwide dissemination of antibiotic resistance, multidrug resistance (MDR) hvKp strains were increasingly reported, namely through horizontal gene transfer between classical-Kp (cKp) and hvKp strains [7]. MDR-hvKp strains can carry virulence and resistance genes on separate plasmids or harbor large hybrid plasmids, encoding both determinants [8]. Alarmingly, carbapenem-resistant hvKp strains (CR-hvKp) are now emerging, causing severe infections, coupled with restricted antibiotic options, leading to elevated morbidity and mortality [4,9]. It is postulated that CR-hvKp can evolve either from hvKp strains through acquiring carbapenemases-encoding plasmids or from carbapenem-resistant cKp by acquiring hypervirulence genes. Worryingly, the presence of a hybrid plasmid carrying both carbapenemase and virulence genes can cause rapid evolution of CR-hvKp from carbapenem susceptible cKP through a single conjugation event [10].

Recently, CR-hvKp has been reported in many regions of Egypt [11,12]. Despite this, there are only a few studies investigating the genomic characteristics of CR-hvKp [11,13,14]. This limited number of studies may be attributed to the relatively recent recognition of CR-hvKp in Egypt, the cost of genomic characterization, and the lack of routine screening for virulence factors in hospitals. Our previous study demonstrated the predominance of MDR-*K. pneumoniae* as the causative agent of VAP in a pediatric ICU (intensive care unit) [12]. A subsequent investigation, involving 53 hvKP isolated from VAP patients admitted to different ICUs at Assiut University Hospitals, revealed that similar PFGE hvKp clones were isolated from diverse ICUs at different times suggesting the likelihood of long-term colonization, as well as the dissemination of hvKp in the hospital environment. On the other hand, different PFGE hvKp clones were found to carry similar virulence and resistance genes, suggesting the presence of endemic plasmids circulating in the hospital environment [11,12]. Thus, an in-depth analysis of plasmids encoding carbapenemase and virulence determinants may provide insight into the evolution of MDR hvKp strains in Egypt and, importantly, provide aid in their control.

Accordingly, in this study, we analyzed the phenotypic and genomic characteristics (i.e., the plasmids, virulence-associated genes, and AMR genes) of two CR-hvKp strains, K22 and K45. Both strains were isolated at the pediatric ICU of Assiut University Hospital, Egypt, in 2015 from patients suffering from VAP. The detection of CR-hvKp, underlines the threat posed by such pathogens in ICU settings and supports the need for strengthened surveillance and more effective screening procedures to identify and isolate hvKp strains.

## 2. Materials and Methods

### 2.1. Bacterial Isolates

The current study is a retrospective study analyzing two clinical CR-hvKP strains (K22 and K45) isolated from the pediatric ICU at Assiut University Hospital, Egypt. Both strains were retrieved from endotracheal aspirates of patients suffering from VAP who were enrolled in a previous study [12]. Strains were categorized as hvKP based on the detection of virulence genes *iucA*, *rmpA*, *rmpA_2_*, *iroN*, and *iroB* [1,2]. They were identified as carbapenem-resistant by determining the minimum inhibitory concentrations (MICs) of imipenem using an E-test. The MIC for ciprofloxacin was also done using an E-test (BioMérieux, Solna, Sweden) and was interpreted according to the Clinical and Laboratory Standard Institute (CLSI) [15].

### 2.2. Antimicrobial Susceptibility Testing

Antimicrobial susceptibility testing was determined through the Kirby–Bauer disk diffusion method [16] using antibiotic disks (ThermoFisher Scientific, Waltham, MA, USA) representing different classes of antibiotics, including penicillin derivatives [amoxicillin (AML10 μg), amoxicillin/clavulanic acid (AMC 20/10 μg), piperacillins (PI 100 μg)], cephalosporins [cefazolin (CZ 30 μg), cefpodoxime (CPD 30 μg), cefoperazone (CPZ 75 μg), ceftriaxone (CTR 30 μg)], aminoglycosides [gentamicin (GE 10 μg), amikacin (AK 30 μg)], tetracycline (TE 30 μg), chloramphenicol (C 30 μg), and trimethoprim sulfonamide (SXT 1.25/23.75 μg). The results were interpreted according to the guidelines stated by CLSI 2023 [15].

### 2.3. String Test

The string test was performed for the detection of the hypermucoviscous phenotype using a standard bacteriologic loop (Fisher Scientific, Loughborough, UK) to generate a viscous string from a single bacterial colony. Strains with a string >5 mm were considered positive for the string test [17].

### 2.4. Whole-Genome Sequencing of Clinical CR-hvKP Strains

The genome sequencing of the two CR-hvKP strains was performed using Illumina sequencing at Microbes NG (https://microbesng.com/ (accessed on 25 April 2025)). Bacterial cultures were preserved in a DNA/RNA Shield cryopreservative (Zymo Research, Irvine, CA, USA) and 45 µL of cell suspension was lysed with 120 μL of TE buffer, containing lysozyme (MPBio, Santa Ana, CA, USA), metapolyzyme (Sigma Aldrich, St. Louis, MO, USA) and RNase A (ITW Reagents, Barcelona, Spain), and incubated for 25 min at 37 °C. Proteinase K (VWR Chemicals, OH, USA) (final concentration 0.1 mg/mL) and SDS (Sigma-Aldrich, St. Louis, MO, USA) (final concentration 0.5% *v*/*v*) were added and incubated for 5 min at 65 °C. Genomic DNA was purified using an equal volume of SPRI beads and resuspended in EB buffer (10 mM Tris-HCl, pH 8.0). Extracted DNA was then quantified with the Quant-iT dsDNA HS (ThermoFisher Scientific, Waltham, MA, USA) assay in an Eppendorf AF2200 plate reader (Eppendorf UK Ltd, Hamburg, UK) and diluted as appropriate.

Genomic DNA libraries were prepared using the Nextera XT Library Prep Kit (Illumina, San Diego, CA, USA), following the manufacturer’s instructions. DNA quantification and library preparations were carried out on a Hamilton Microlab STAR automated liquid handling system (Hamilton Bonaduz AG, Bonaduz, Switzerland). Sequencing was performed with the Illumina NovaSeq 600 (Illumina, San Diego, CA, USA), using a 250 bp paired-end protocol. Adapter trimming of Illumina reads was done using Trimmomatic 0.30, with a sliding window quality cutoff of Q15 [18], producing 1,226,225 trimmed reads for K22 and 786,858 for K45. Genome assembly was achieved using Spades version 3.7 [19] and contigs were annotated using Prokka 1.11 [20]. This Whole Genome Shotgun project has been deposited at the National Center for Biotechnology Information (NCBI) GenBank as BioProject PRJNA1079765, with the sequence data for clinical CR-hvKP (K22 and K45) under accession numbers JBAMJT000000000 and JBAMJS000000000, respectively.

### 2.5. Genomic Analysis

Draft genomes were visualized using Artemis [21], and comparisons between genomes were examined using the Proksee website (https://proksee.ca/about (accessed on 25 April 2025)) [22], the Basic Local Alignment Search Tool (BLAST) at NCBI (https://blast.ncbi.nlm.nih.gov/Blast.cgi (accessed on 25 April 2025)) and the Artemis Comparison Tool (ACT) [23]. Representations of genome organization were drawn using the Proksee Server [22] and ACT [23]. Sequence types were determined using MLST 2.0 [24], and the presence of plasmids was determined by detecting plasmid replicons using PlasmidFinder 2.1 [25] with the online software from the Center for Genomic Epidemiology (CGE) (http://www.genomicepidemiology.org/ (accessed on 25 April 2025)). Antibiotic resistance genes were detected using ResFinder 3.2 at CGE [26] and using the ResFinder (22 March 2024) and PointFinder (8 March 2024) databases at the standard setting of 90% and 60% for threshold and length ID, respectively. Resistance gene identity was further corroborated by interrogating the Prokka annotation of each isolate [20], BLAST analysis at NCBI and the Comprehensive Antibiotic Resistance Database (CARD) using Proksee [22,27]. Kaptive Web (https://kaptive-web.erc.monash.edu/ (accessed on 25 April 2025)) was used for the determination of capsule type and O-antigen locus type [28]. Furthermore, heavy metal-resistance genes were detected using the BIGSdb-Kp database at the Institut Pasteur (https://bigsdb.web.pasteur.fr/index.html (accessed on 25 April 2025)). The Virulence Factor Database (http://www.mgc.ac.cn/VFs/main.htm (accessed on 25 April 2025)) was used to screen for virulence genes [29]. Insertion sequences and bacteriophage were identified using ISfinder (https://www-is.biotoul.fr/index.php (accessed on 25 April 2025)) [30] and PHASTER (https://phaster.ca/ (accessed on 25 April 2025)) [31], respectively.

## 3. Results

### 3.1. Bacterial Isolation and AMR

The two clinical CR-hvKP isolates, K22 and K45, were isolated from children with VAP admitted to the pediatric ICU at Assiut University Children’s Hospital in 2015 [12]. Phenotypic analysis indicated that these two isolates were highly resistant to almost all tested antibiotics with both isolates resistant to the front-line carbapenem antibiotic, imipenem (Table 1). Thus, K22 is classified as possessing extensive drug resistance phenotype (XDR), while K45 exhibited an MDR phenotype [32]. However, it is of note that both isolates were still susceptible to tetracycline and chloramphenicol, with K45 additionally susceptible to fluoroquinolones. Furthermore, both strains were negative for the string test, which is used to define the hypermucoviscous phenotype (Table 1) [17].

### 3.2. Genomic Characterization of CR-hvKP Isolates

To understand more about each isolate, the genomes of K22 and K45 were sequenced using Illumina short-read sequencing, which revealed that they possessed respective genome sizes of 5,862,489 bp and 5,741,936 bp, with both carrying multiple plasmid replicons (Table 2). Comparison of each strain’s genome with the chromosome of the archetype *K. pneumoniae* reference strain, MGH 78587 (CP000647.1) [33], revealed that both differed from MGH 78578 in their capsule and O-antigen locus, along with other chromosomal positions, which included bacteriophage, sugar utilization operons and secretions systems (Figure 1). Both isolates possessed an O1v1 O-antigen loci, whilst K22 was capsular type K30 and K45 capsular type K2 (Table 2). Note that K2 capsule is often associated with hypervirulence in *K. pneumoniae* [34]. Sequence typing indicated that K22 was ST383 and K45 was ST14, both of which have been designated as high-risk *K. pneumoniae* clonal complexes and isolated in Egypt before [11,14,35].

### 3.3. Antimicrobial Resistance Genes

Analysis of the antimicrobial resistance genes carried via K22 and K45 agreed with their antimicrobial susceptibility profiles (Table 1), with each strain carrying genes associated with resistance to aminoglycoside, β-lactam, trimethoprim sulfonamide, macrolide, and fosfomycin antibiotics (Table 3). Importantly, both strains carried the carbapenemase gene *bla*_OXA-48_, whilst K22 also possessed *bla*_NDM-1_, explaining the resistance of both strains to imipenem (Table 1). K22 also possessed the *aac*(6′)-Ib-cr gene, which affords protection against both aminoglycoside and fluoroquinolone antibiotics, explaining this isolate’s resistance to ciprofloxacin (Table 1). In addition to acquired AMR genes, each strain also carried various point mutations in chromosomal genes such as *gyrA*, *acrR*, and *parC*, which can influence resistance to fluoroquinolones, and alternations in the genes encoding outer membrane porins *ompK36* and *ompK37*, which are associated with carbapenem resistance (Table 3) [26].

### 3.4. Virulence Determinants

Whole-genome sequence analysis also identified many virulence-associated genes within our isolates (Table 3). These included genes involved in adhesion and biofilm formation, such as those encoding the *E. coli* common pilus (*ecpRABCDE*), mannose-sensitive type I fimbriae (*fimABCDFGH*), and *Klebsiella* mannose-insensitive type III fimbriae (*mrkABCDF*, *mrkHIJ*). Both isolates carried many genes for siderophore production and uptake, which included those for enterobactin production (*fepA-entD*, *entF-fes*, *fepDGC*, *entS*, *fepB*, *entCEBAH*), the aerobactin gene cluster and receptor (*iucABCD*, *iutA*), and the salmochelin siderophore esterase and receptor (*iroE*, *iroN*). Additionally, K45 also carried the *kfuABD* iron uptake genes and both strains carried the *rmpA* and *rmpA2* transcription regulators genes, which are commonly found in hvKP strains. It is of note that, in both strains, *rmpA2* was a pseudogene, due to a frame shift within the open reading frame.

### 3.5. Plasmid-Associated Antimicrobial Resistance and Virulence Genes

Typical of clinically isolated CR-hvKP strains, K22 and K45, carried multiple plasmid replicons (Table 2). Of particular note is that both K22 and K45 carry identical IncFIB (pNDM-Mar) and IncHI1B (pNDM-Mar) replicons (Appendix A). BLASTn analysis of contigs carrying the IncFIB (K22 contig 15: 63,461 bp; K45 contig 25: 62,414 bp) and the IncHI1B replicons (K22 contig 23: 68,431 bp; K25 contig 12: 90,554 bp) indicated that they possess a high degree of similarity to plasmid pKP-1PI_HIB-FIB (CP071028.1) (Figure 2; Appendix A). Plasmid pKP-1PI_HIB-FIB (also known as pVIR-147Tu [37]) was isolated in Tuscany, Italy, during an outbreak of an NDM-1-producing *K. pneumoniae* in 2018. The plasmid was found to be a multi-replicon plasmid formed through the fusion of IncFIB and IncHI1B plasmid backbones that was capable of conjugative transfer. Additionally, pKP-1PI_HIB-FIB carries multiple AMR genes (*aph*(3′)-Ia, *armA*, *dfrA5*, *mphE*, *msrE*, *mphA*, *sul1*, and *sul2*,), hypervirulence-associated genes (*rmpA*, *iucABCD-iutA*, and a *rmpA2* pseudogene), tellurium tolerance genes (*terABCDEWXYZ*) and various transfer (*tra*) genes (e.g., *traD* and *traN*) (Figure 2; Appendix A) [37]. As these genes are similarly organized in the draft genomes of K22 and K45 (Appendix A), our results suggest that both strains harbor a similar conjugative IncHI1B/FIB hybrid virulence plasmid, which we term pK22-Vir and pK45-Vir.

The dissemination of carbapenemase genes in Gram-negative bacteria has been primarily through the transfer of conjugative plasmids. Therefore, we sought to determine which K22 and K45 plasmid replicons harbor these resistance determinants. Our analysis indicates that *K. pneumoniae* K22 carries an IncL replicon (K22 contig 30: 41,120 bp), whilst K45 a related IncM1 replicon (K45 contig 14: 66,715 bp), both of which have been implicated in the carriage of specific carbapenemase genes (Table 2; Appendix A) [38]. Blastn analysis of the K22 IncL-containing contig, which also carries the *aph*(3″)-Ib and *aph*(6)-Id AMR genes, indicated that it was identical to plasmid pDT1 (CP019078.1), carried via *K. pneumoniae* strain DT1, isolated in Hamburg, Germany in 2015 during an ICU outbreak (Appendix A) [39]. Like K22, strain DT1 was sequence type ST383, and pDT1 carried *bla*_OXA-48_ and the extended-spectrum beta-lactamase (ESBL) gene, *bla*_CTX-M-14_ (Figure 3A). An analysis of the IncM1 replicon from K45, which carries *aph*(3′)-VIb, *aph*(3″)-Ib, *aph*(6)-Id, and *bla*_CTX-M-14b_ AMR genes, indicated that it was identical to plasmid pOXA48-Pm (KP025948.1), a broad host range plasmid that was isolated from *Proteus mirabilis* strain Pm-Oxa48 from Gaza in 2012 (Appendix A) [40]. Plasmid pOXA48-Pm also carries multiple AMR genes, including *bla*_OXA-48_ and the *bla*_CTX-M-14b_ ESBL (Figure 3B) [40]_._ Thus, our analysis suggests that *bla*_OXA-48_ carbapenemase genes, carried via K22 and K45, are likely located on IncL and IncM1 plasmids, which we term pK22-OXA48 and pK45-OXA48, respectively (Figure 3). As these plasmids carry multiple *tra* genes, it is possible that they might be conjugative in nature (Figure 3; Appendix A).

In addition, *K. pneumoniae* K22 also carries the *bla*_NDM-1_ carbapenemase gene. Our analysis indicated that the K22 contigs carrying both *bla*_NDM-1_ (K22 contig 34: 30,026 bp) and the IncFII(Yp) replicon (K22 contig 27: 55,018 bp) were very similar to those of the *Klebsiella michiganensis* plasmid, pK518_NDM1 (CP023187.1) (Figure 4; Appendix A), which was isolated from Zhejiang, China, in 2017 [41]. Thus, due to similarities observed between pK518_NDM1, we propose that the *bla*_NDM-1_ carbapenemase of *K. pneumoniae* strain K22 is carried on a similar conjugative IncFII (YP) plasmid, which we term pK22-NDM1.

*K. pneumoniae* strain K22 also carries IncFIB(pQil) and Col440II plasmid replicons, whilst strain K45 carried Col(pHAD28) and Col440I replicons (Table 2; Appendix A). Of note is that the K22 IncFIB(pQil) replicon (K22 contig 38: 18,962 bp) was similar to plasmid pKPN4 (CP000649.1) from the archetypal *K. pneumoniae* strain MGH 78587 [33], though this was restricted to the surrounding replicon. As none of these replicon contigs carried virulence or AMR genes, their contribution to the virulence and AMR profiles of K22 and K45 is unclear.

## 4. Discussion

Carbapenem-resistant hypervirulent *K. pneumoniae* is considered a worldwide threat due to its ability to cause life-threatening infections with limited therapeutic options [4,9]. In the current study, we have investigated the phenotypic and genomic characteristics of two clinical CR-hvKp isolates, K22 and K45, recovered from VAP patients admitted to the pediatric ICU at Assiut University Hospitals in 2015. Both isolates were identified as hypervirulent based on the presence of the virulence genes; *iucA*, *rmpA*, and *rmpA_2_* (Table 3) [42]. In addition to specific hypervirulence genes, each strain carried other *K. pneumoniae* virulence-associated genes, including the *mrkABCDF* type III fimbriae operon, as well as genes involved in the production and/or uptake of enterobactin and salmochelin siderophores (Table 3). Moreover, strain K45 carried the *kfuABD* iron scavenging system and was found to be of capsular type K2, which is frequently associated with hypervirulent strains [34,43,44]. Both isolates carried the O1 O-antigen locus suggested to be the most common O-antigen locus detected in hvKp strains [43]. Interestingly, both strains were negative for the string test, most probably due to the presence of a frameshift mutation in *rmpA_2_* (Table 1) [8].

AMR profiling indicated that K22 exhibited an XDR phenotype, whilst K45 was MDR (Table 1). The AMR phenotype was consistent with the presence of many acquired resistance genes and specific chromosomal point mutations detected in their draft genomes (Table 3). Furthermore, both strains carried multiple plasmid replicons, which are localized on contigs containing multiple AMR genes, as well as *tra* genes (Table 2; Appendix A). The detection of numerous resistance genes located on conjugative plasmids suggests that these KP isolates may serve as potential reservoirs for AMR dissemination in the pediatric ICU setting. While no conjugation assays were performed in this study, similar plasmid-mediated transmission has been reported in previous studies [45,46].

Plasmids acquire resistance genes through small mobile elements, such as integrons and transposons, and contribute to resistance expansion through horizontal transmission [47]. In the current study, K22 and K45 were found to carry several resistance/virulence plasmids. Both strains were resistant to the carbapenem imipenem and carried known carbapenemase genes (Table 1 and Table 3). Our analysis indicated that both K22 and K45 carried an IncL/M plasmid (i.e., pK22-OXA48 and pK45-OXA48, respectively) encoding *bla_OXA-48_* carbapenemase, as well as aminoglycosides resistance genes (e.g., *aph*(3″)-Ib and *aph*(6)-Id)) (Figure 3; Appendix A). The K22 IncL plasmid pK22-OXA48 was similar to *K. pneumoniae* pDT1, which was isolated during a German ICU outbreak in 2015 (Figure 3A) [39], whilst the IncM1 plasmid pK45-OXA48, carried by K45, was very similar to *P. mirabilis* plasmid pOXA48-Pm, isolated in 2012 from Gaza (Figure 3B) [40]. Furthermore, K22 also carries the *bla*_NDM-1_ carbapenemase gene on an IncFII(Yp) plasmid (i.e., pK22-NDM1) (Figure 4; Appendix A), which is very similar to *K. michiganensis* plasmid pK518_NDM1 that was isolated in China in 2017 [41]. The presence of similar plasmids among different Gram-negative bacilli could suggest that these plasmids are transmitted by conjugation between different species.

The co-localization of MDR and virulence genes on a single plasmid is a catastrophic incident, likely establishing a speedy evolutionary path for MDR-hvKp from susceptible cKp through a sole conjugation incident [48]. Importantly, both K22 and K45 carried a hybrid IncHI1B/FIB plasmid, which contained both AMR and virulence genes (i.e., *rmpA* and *iucABCD-iutA*). As these genes are similarly organized in the draft genomes of K22 and K45 (Appendix A), it is suggested that both strains harbor similar IncHI1B/FIB hybrid plasmids (i.e., pK22-Vir and pK45-Vir). Moreover, 39/53 of the hvKp strains isolated later in a second study (2017–2019), harbored both *rmpA* and *iucA* genes, suggesting that these 39 hvKp strains might also have carried this hybrid IncHI1B/FIB plasmid [11,12].

Sequence typing indicated that K22 belongs to ST383, whilst K45 was ST14 (Table 1), and both STs are considered as high-risk *K. pneumoniae* clones. ST14 has been linked to hypervirulence and carbapenemase carriage, being the first *K. pneumoniae* sequence type in which the *bla*_NDM-1_ carbapenemase gene was identified [35,49,50,51,52]. On the other hand, ST383 is an evolving high-risk clonal complex that was first detected in Greek hospitals in 2009 [53]. Since then, it has disseminated throughout the world to many countries, including Egypt where CR-hvKp ST383 isolates have been detected in different cities [11,14,54]. It is noteworthy that a hvKp ST383 *K. pneumoniae* strain isolated in Alexandria (Egypt) in 2021 and another CR-hvKp ST11 strain isolated in Cairo in 2017, both harbored an IncHI1B/FIB virulence plasmid and an IncL-carrying *bla*_OXA-48_ plasmid, similar to those of K22 (Appendix A) [14,55].

The IncHI1B/FIB hybrid plasmids found in K22 and K45 are similar to conjugative plasmid pKP-1PI_ HIB-FIB, isolated during an Italian *K. pneumoniae* outbreak in 2018 (Figure 2; Appendix A) [37] and very similar plasmids have been isolated from *K. pneumoniae* strains in the UK, Poland, Russia, and Egypt (Appendix A) [55,56,57,58,59]. The earliest documented occurrence of this specific type of hybrid IncHI1B/FIB plasmid is pKpvST147L (CM007852.1), which was isolated from a patient in the UK in 2016 (Appendix A) [56]. As this patient’s country of residence was Egypt, this could suggest that their infection with *K. pneumoniae* occurred whilst the patient was in Egypt. Accordingly, as strain K22 was isolated in 2015, it could possibly be considered to be the “ground zero” strain, being the first isolate to harbor this specific hybrid IncHI1B/FIB AMR/virulence plasmid. Therefore, further large-scale studies are desperately required to reveal the prevalence of such potentially devastating plasmids and to better understand their role in the spread of MDR-hvKp strains in Egypt.

## Figures and Tables

**Figure 1 microorganisms-13-01058-f001:**
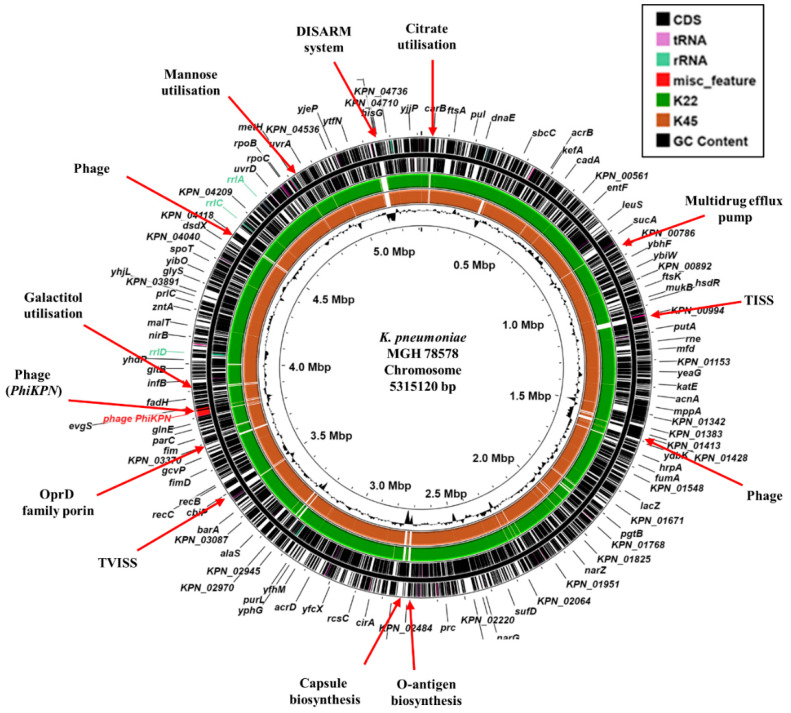
Comparison of the *K. pneumoniae* MGH 78578 chromosome with the draft genomes of *K. pneumoniae* isolates K22 and K45. The figure shows the comparison of the *K. pneumoniae* MGH 78578 chromosome with the draft genomes of *K. pneumoniae* K22 and K45, using ProkSee [22]. The two outer rings display the genes/CDS of the *K. pneumoniae* MGH 78578 chromosome (CP000647.1) with selected features indicated [33]. The green and brown rings depict the BLAST results when the contigs from the draft genome sequences of K22 and K45 are compared with the MGH 78578 chromosome with shaded regions indicating homology. The innermost ring displays the GC content for the *K. pneumoniae* MGH 78578 chromosome. Abbreviations: TISS, Type I secretion system; TVISS, Type VI secretion system; DISARM, Defense Island System Associated with Restriction-Modification [36].

**Figure 2 microorganisms-13-01058-f002:**
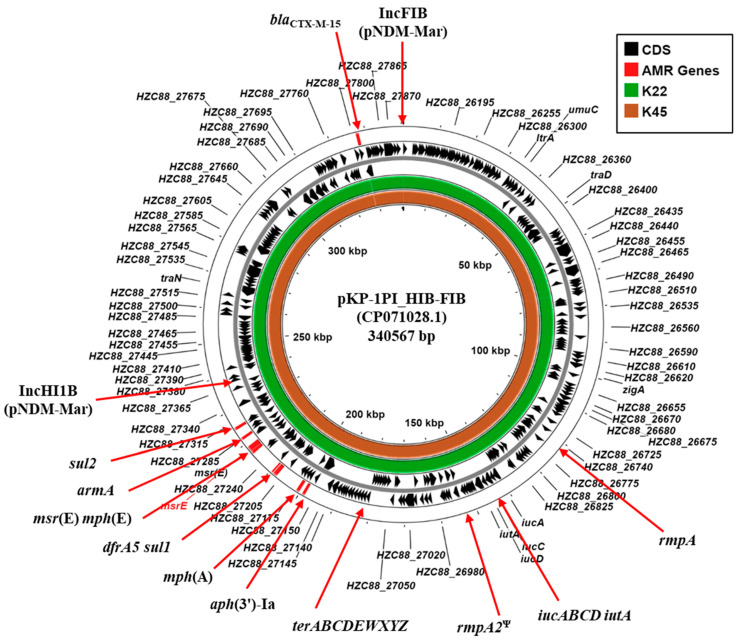
Comparison of the *K. pneumoniae* plasmid pKP-1PI_HIB-FIB with the draft genomes of *K. pneumoniae* isolates K22 and K45. The figure shows the comparison of *K. pneumoniae* plasmid pKP-1PI_HIB-FIB (also known as pVIR-147Tu [37]) with contigs from the draft genomes of *K. pneumoniae* K22 and K45, using ProkSee [22]. The genes/CDS of pKP-1PI_HIB-FIB (CP071028.1) is displayed with selected features indicated [37]. The green and brown rings depict the BLAST results when the contigs from the draft genome sequences of K22 and K45 are compared with pKP-1PI_HIB-FIB. ^Ψ^ pseudogene.

**Figure 3 microorganisms-13-01058-f003:**
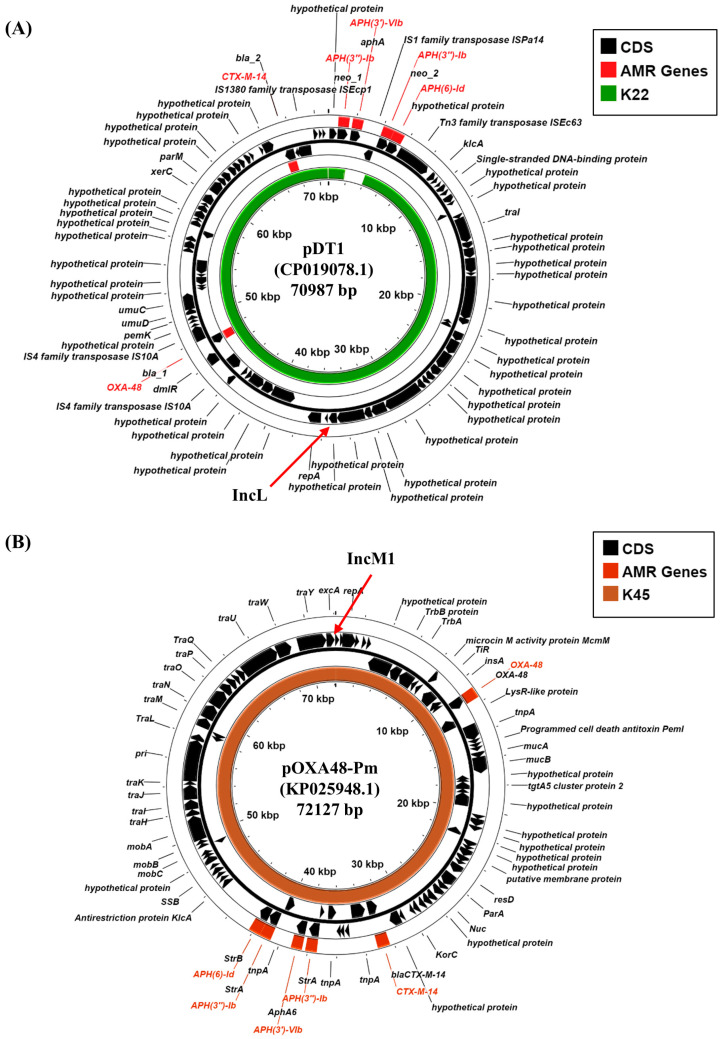
Analysis of the IncL and IncM plasmid replicons carried via *K. pneumoniae* isolates K22 and K45. (**A**) The panel shows a comparison of *K. pneumoniae* IncL plasmid pDT1 (CP019078.1) [39] with the draft genome of *K. pneumoniae* K22, using ProkSee [22]. (**B**) The panel shows a comparison of *P. mirabilis* IncM1 plasmid pOXA48-Pm (KP025948.1) [40] with the draft genome of *K. pneumoniae* K45, using ProkSee [22]. The green and brown rings depict the BLAST results when the contigs from the draft genome sequences of K22 and K45, respectively, are compared with each relevant plasmid.

**Figure 4 microorganisms-13-01058-f004:**
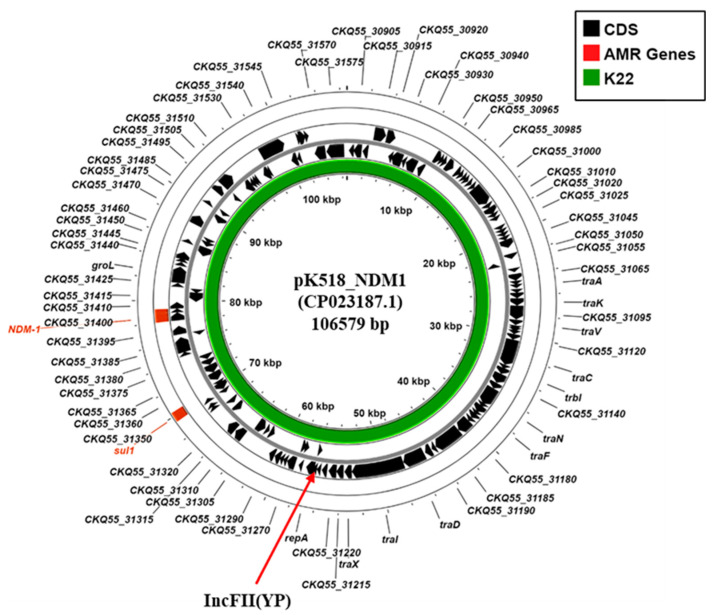
Analysis of the IncFII(YP) replicon carried via *K. pneumoniae* isolates K22. The figure shows the comparison of *K. michiganensis* IncFII(YP) plasmid pK518_NDM1 (CP023187.1) [41] with the draft genome of *K. pneumoniae* K22, using ProkSee [22]. The green rings depict the BLAST results when the contigs from the draft genome of K22 are compared to pK518_NDM1.

**Table 1 microorganisms-13-01058-t001:** Characterization and antimicrobial susceptibility testing of clinical CR-hvKP isolates.

Strain ID.	K22	K45
Date of isolation	10 January 2015	12 May 2015
ICU isolation	Pediatric	Pediatric
Type of infection	VAP	VAP
MIC IMP	12 (R)	4 (R)
MIC CIP	>32 (R)	0.047 (S)
Resistance phenotype *	XDR ^1,2,3,6,7,8^	MDR ^1,2,3,7,8^
String test	Negative	Negative

Abbreviations: ICU, intensive care unit; VAP, ventilator-associated pneumonia; MIC, minimum inhibitory concentration; IMP, imipenem; CIP, ciprofloxacin; R, resistant; S, sensitive; MDR, multi-drug resistance; XDR, extensive drug resistance. * Key: ^1^, penicillin resistance; ^2^, cephalosporin resistance; ^3^, carbapenem resistance; ^6^, fluoroquinolone resistance; ^7^, trimethoprim sulfonamide resistance; ^8^, aminoglycoside resistance.

**Table 2 microorganisms-13-01058-t002:** Genetic characteristics of CR-hvKP clinical isolates from Assiut, Egypt.

Strain ID.	K22	K45
Genome size	5,862,489 bp	5,741,936 bp
Number of Contigs	151	101
Genes (CDs)	5580	5344
GC%	56.54%	56.65%
MLST	ST383	ST14
Capsular type	K30	K2
O type	O1/O2v1 Type O1	O1/O2v1 Type O1
Plasmid replicons	IncFIB (pNDM-Mar) IncHI1B (pNDM-Mar) IncFIB (pQil) IncFII (Yp) IncL, Col440II	IncFIB (pNDM-Mar) IncHI1B (pNDM-Mar) IncM Col(pHAD28) Col440I

**Table 3 microorganisms-13-01058-t003:** Antimicrobial resistance, heavy metal resistance and virulence-associated genes detected in the K22 and K45 CR-hvKP clinical isolates.

Strain	K22	K45
Acquired AMR gene ^1^	**Amg^R^:** *aadA1*, *aph*(3′)-Ia, *aph*(3″)-Ib, *aph*(6)-Id, *armA*, *rmtC* **Amg^R^/Flq^R^:** *aac*(6’)-Ib-cr **βL^R^:** *bla*_CTX-M-14b_, ***bla*_NDM-1_**, *bla*_OXA-9_, ***bla*_OXA-48_**, *bla*_SHV-26_ **TS^R^:** *dfrA5*, *sul1*, *sul2* **Mcl^R^:** *mph*(A), *mph*(E), *msr*(E) **Fos^R^:** *fosA*	**Amg^R^:** *aph*(3′)-VIb *aph*(3′)-Ia, *aph*(3″)-Ib, *aph*(6)-Id, *armA* **βL^R^:** *bla*_CTX-M-14b_, ***bla*****_OXA-48_**, *bla*_SHV-28_ **TS^R^:** *dfrA5*, *sul1*, *sul2* **Mcl^R^:** *mph(A)*, *mph(E)*, *msr(E)* **Fos^R^:** *fosA*
Chromosomal point mutations associated with AMR	***gyrA*:** S83F, D87N ***parC*:** S80I ***ompK37*:** I70M, I128M ***acrR*:** P161R, G164A, F172S, R173G, L195V, F197I, K201M	***ompK36*** N49S, L59V, L191S, F207W, A217S, N218H, D224E, L228V, E232R, T254 ***ompK37*:** I70M, I128M ***acrR*:** P161R, G164A, F172S, R173G, L195V, F197I, K201M
Virulence genes	**Adhesion/Biofilm formation** *mrkABCDF*, *mrkHIJ*, *fimABCDFGH*, *ecpRABCDE* **Iron acquisition** *iucABCD iutA*, *fepA-entD*, *entF-fes*, *fepDGC*, *entS*, *fepB entCEBAH* *iroE*, *iroN* **Virulence regulation** *rmpA*, *rmpA2* ^Ψ^	**Adhesion/Biofilm formation** *mrkABCDF*, *mrkHIJ*, *fimABCDFGH*, *ecpRABCDE* **Iron acquisition** *iucABCD iutA*, *fepA-entD*, *entF-fes*, *fepDGC*, *entS*, *fepB entCEBAH* *iroE*, *iroN* *kfuABC* **Virulence regulation** *rmpA*, *rmpA2* ^Ψ^
Heavy metal resistance	*cusRS*, *cusCFBA*, *terABCDEWXYZ*	*cusRCFBA*, *terABCDEWXYZ*

^1^ Abbreviations: Amg^R^, aminoglycoside resistance; Flq^R^, fluoroquinolone resistance; βL^R^, β-lactam resistance; TS^R^: trimethoprim sulfonamide resistance; Mcl^R^: macrolide resistance; Fos^R^, fosfomycin resistance. ^Ψ^ pseudogene.

## Data Availability

All genome sequence data and assemblies have been deposited at NCBI GenBank under BioProject ID PRJNA1079765. The assembled and annotated genomes of *Klebsiella pneumoniae* strains K22 and K45 have been deposited with the accession numbers JBAMJT000000000 and JBAMJS000000000, respectively.

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
