# Peer review of "Genome Characterization of Carbapenem-Resistant Hypervirulent Klebsiella pneumoniae Strains, Carrying Hybrid Resistance-Virulence IncHI1B/FIB Plasmids, Isolated from an Egyptian Pediatric ICU"

_microorganisms, 2025, doi:10.3390/microorganisms13051058_

Round 1
Reviewer 1 Report
Comments and Suggestions for Authors
This article reported the genomic characterization of two CR-hvKp isolates in Egypt. The article was purely descriptive, a bit long, over-discussed, and had no comparison to available ST383 and ST14 Kp data. The level of novelty was low. the genomic description and comparison was a bit sloppy. Many sentences had no reference and lack supporting evidence.
Questions and comments:
p1, line 32, last sentence in abstract was unsupported. There was no evidence in the manuscript to support such statement. No genetic analysis performed.
p1, line 38, the first sentence about the hvKp was not that accurate. hvKP has been around for a long time. Do you mean specific STs? This sentence was too broad and out of context.
p2, line 81, These two isolates were from what kind of specimen? Did they cause invasive infection?
p2, line 82. Please add a reference to define hvKp.
p3, line 102. These were not complete genomes, based on short reads only. Remove "complete"
p3, line 124, The genomes were not made available to public.
p5, Table2, plasmids were inferred from plasmidfinder based on contigs. So there was no complete plasmids characterization?
p7, line 210, these statements were misleading. Only the partial contigs that bear signature of Inc replicons were identical. You need to report the length of these contigs. They were not complete plasmids.
p8, Figure 2, these are contigs alignment to the complete plasmid sequence from NCBI, not the comparison of plasmids. No full, complete plasmids were characterized based on Illumina reads.
p8-9, the entire results/discussion related to plasmids, replicons were over-stated. These are short reads assemblies, and no complete plasmids were reported except the contigs that bear OXA-48 families. All the potential conjugation or spread were only prediction. The discussion section on plasmids could be shortened a lot.
p10 Figure 3, contig number need to be included in the diagrams
p12, line 316-317, tune down this sentence and reference other work. No conjugation experiment was performed in this study. And there was no data on conjugation efficiency.
Many studies on Kp ST383 and ST14 have been published. e.g. https://pubmed.ncbi.nlm.nih.gov/38447323/; https://pubmed.ncbi.nlm.nih.gov/38606647/ ; https://pubmed.ncbi.nlm.nih.gov/37310284/; Can you perform phylogenetic analysis with these published data?
p12, line 328-329, why? how? there was no data presented to support this statement.
p12, line 341, ST383 isolates were different genetically. There may be no clonal expansion. Plasmid horizontal transfer has to be in context. Which plasmid?
p12-13, line 357-363, lack of data supporting "close genetic relationship" plasmids in this study. discussion was too superficial and hypothetical.
Author Response
Please see attached "Replies to Reviewer's Comments" Document.

Reviewer 2 Report
Comments and Suggestions for Authors
The manuscript “Genome characterization of carbapenem resistant hypervirulent Klebsiella pneumoniae strains, carrying hybrid resistance-virulence IncHI1B/FIB plasmids, isolated from an Egyptian pediatric ICU” by Hammad et al. characterized two carbapenem-resistant hypervirulent Klebsiella pneumoniae (CR-hvKp) strains, K22 and K45, isolated from pediatric ICU patients with ventilator-associated pneumonia (VAP) in Egypt, revealing that K22 exhibited extensive-drug resistance (XDR) while K45 showed multidrug resistance (MDR). Both strains carried genes associated with hypervirulence and multiple plasmids, including a hybrid IncHI1B/FIB plasmid carrying both antimicrobial resistance (AMR) and virulence genes, suggesting its role in disseminating both traits within the hospital environment, with K22 potentially representing one of the earliest carriers of this hybrid plasmid in Egypt.
The article demonstrates strong presentation quality. Minor refinements in some sections are suggested:
Introduction:
Lines 38-39: “Hypervirulent K. pneumoniae (hvKp) is a newly emerging pathogen evolved through the acquisition of potent virulence determinants”. Before this sentence the authors could mention some aspects about the biology of Klebsiella pneumoniae.
Lines 61-62: “Recently, CR-hvKp have been reported in many regions of Egypt [11, 12]. Despite that, there are only a few studies investigating the genomic characteristics of CR-hvKp”. What is the reason for this? Is this due to the lack of characterization of them? The authors could write a sentence about it.
Lines 73-76: “Accordingly, in this study, we analyzed the phenotypic and genomic characteristics (i.e., the plasmids, virulence-associated genes, and AMR genes) of two CR-hvKp strains, K22 and K45. Both strains were isolated at the pediatric ICU (intensive care unit) of Assiut University Hospital, Egypt, in 2015 from patients suffering from VAP”. Here the authors could give more importance to the results obtained in this study and how they could be used in the implementation of public health policies in the hospital.
Materials and Methods:
Lines 84-85: “They were identified as carbapenem resistant by determining the minimum inhibitory concentrations (MICs) of imipenem using an E-test”. The authors could explain why imipenem was chosen as the carbapenem antibiotic for the MIC test.
Lines 119-120: “Adapter trimming of Illumina reads was done using Trimmomatic 0.30 with a sliding window quality cutoff of Q15”. Why was a quality score of 15 chosen when the standard quality score used for Illumina reads is 30? Very low quality values ​​in Illumina can lead to the introduction of errors in the assembly.
Lines 128-132: “Sequence types were determined using MLST 2.0 [22], antibiotic resistance genes were detected using ResFinder 3.2 [23], and the presence of plasmids was determined by detecting plasmid replicons using PlasmidFinder 2.1 [24] with the online software from the Center for Genomic Epidemiology (CGE) (http://www.genomicepidemiology.org/)”. The authors should clarify why they chose Resfinder exclusively for resistance gene identification, and not other programs like RGI, AMRFinderPlus, or Abricate for result validation. Additionally, they should specify the database used in the search, along with the identity and coverage percentage thresholds employed.
The methodology for the comparative genomics analyses presented in the results section is missing. Please include this information.
Results
Line 155: “...revealed that both differed from MGH 78578…”. ¿MGH78578 or MGH 78587? Please correct.
Line 182: “Table 2. Genetic characteristics of CR-hvKP clinical isolates from Assiut, Egypt”. The authors could add a new line in the table indicating the number of clean reads used in each assembly.
Lines 205-206: “Table 3. Antimicrobial resistance, heavy metal resistance and virulence-associated genes detected in the K22 and K45 CR-hvKP clinical isolates”. The authors could add a new line in the table indicating the number of clean reads used in each assembly. The authors could add in this table the values of identity and coverage percentage identified in each gene. Optionally, this information could be placed in a supplementary table.
Lines 212-214: “BLASTn analysis of contigs carrying these replicons indicated that they possess a high degree of similarity to plasmid pKP-1PI_HIB-FIB (CP071028.1)”. The methodology doesn't indicate which programs were used in this analysis. Please include them in the corresponding section.
Discussion
Lines 274-275: “In addition to specific hypervirulence genes, each strain carried other K. pneumoniae virulence-associated genes…”. The authors are recommended to relate the presence of the virulence genes reported in each genome with the clinical manifestations of the patients from which they were isolated.
Lines 292-294: “Interestingly, both strains were negative for the string test, most probably due to the presence of a frameshift mutation in rmpA2”. The authors could further explore the biological significance of the presence of this pseudogene.
Lines 296-297: “The AMR phenotype was consistent with the presence of many acquired resistance genes and specific chromosomal point mutations detected in their draft genomes”. The authors could explore and explain more deeply the importance of the presence of these point mutations.
Comments on the Quality of English LanguageThe manuscript requires revision by a native language specialist to address grammatical errors and improve paragraph structure.
Author Response

(The authors gave the same response as above.)

Round 2
Reviewer 1 Report
Comments and Suggestions for Authors
These two samples were from Endotracheal aspirates. What was the clinical significance of hvKp in respiratory infection?
Otherwise, allquestions have been addressed.
Reviewer 2 Report
Comments and Suggestions for Authors
I have reviewed the resubmission of the manuscript entitled "Genome characterization of carbapenem resistant hypervirulent Klebsiella pneumoniae strains, carrying hybrid resistance-virulence IncHI1B/FIB plasmids, isolated from an Egyptian pediatric ICU". The authors answered in a satisfactory way the points that I have addressed in the first review. Thus this new version of the manuscript can be accepted for publication.